# Evaluating E-Government Development among Africa Union Member States: An Analysis of the Impact of E-Government on Public Administration and Governance in Ghana

Bernard John Tiika [1,2,3,*] , Zhiwei Tang [1,2,*], Jacob Azaare [4] , Joshua Caleb Dagadu [5]
and Samuel Nii-Ayi Otoo [3]

1 School of Management and Economics, University of Electronic Science and Technology of China,
Chengdu 610054, China
2 Center for West Africa Studies, University of Electronic Science and Technology of China,
Chengdu 610054, China
3 Central Administration, University for Development Studies, Tamale P.O. Box TL1350, Ghana
4 School of Computing and Information Sciences, C. K. Tedam University of Technology and Applied Sciences,
Navrongo P.O. Box 24, Ghana; jazaare@cktutas.edu.gh
5 Department of Information Technology Education, Akenten Appiah-Menkah University of Skills Training and
Entrepreneurial Development, Kumasi P.O Box 1277, Ghana
* Correspondence: btiika@uds.edu.gh (B.J.T.); tangzw@uestc.edu.cn (Z.T.)

**Abstract:** The adoption of e-government promises efficiency in the delivery of government services to citizens across various sectors of the economy. Due to this, most Global North countries have advanced in the deployment of e-government for improving public-service delivery. Unfortunately, most African countries, including Ghana, are still lagging in e-government development. This study examined e-government development across African Union member states. It explored the role of e-government in the reform of public administration and governance, focusing on Ghana as a case study. Using a mixed-method approach, the study analyzed secondary data of key e-government indicators using the TOPSIS method. This helped underscore the transformative impact on public administration and governance by using primary data via interviews. The results show advanced progress in some African countries, including Ghana, due to aligned digital strategies with national policies. Also, technology integration is evident in Ghana's public sector and is reshaping public administration and governance. The study recommends that to achieve the long-term sustainability of these advancements, interagency collaboration and data-sharing mechanisms between the public and private sectors should be strengthened, while all forms of silos should be broken to promote the delivery of services. This study enhances public-service delivery by identifying areas needing both improvement and allocation of resources for shaping e-government policy development.

**Keywords:** e-government; public administration; good governance; African Union; TOPSIS; Ghana





## 1. Introduction

There has been tremendous recognition across the globe that effective public-sector governance involves the application of e-government to achieve efficiency and improve the delivery of government services to individuals and organizations. Governments across the globe are investing and committing significant financial resources to the development of e-government and ICT. As of 2022, global IT expenditure amounted to over USD 548 billion, with an expected 7.8% increment to about USD 590 billion in 2023 [1,2]. For the Middle East and Africa, the estimated IT expenditure was about USD 105.3 billion in 2016, and it is expected to increase to USD 155.8 billion by 2023, with a compound annual growth rate of 6.3% from 2018 to 2023 [3]. The evolution of service provision from traditional "brick-and-mortar" systems to advanced technology applications highlights the transformative journey of electronic government (e-government). Progressing from basic websites to sophisticated

web browsers and mobile apps, e-government holds promising potential to revolutionize public administration, governance, and sustainable development [4–7].

E-government refers to the application of ICTs to deliver government services to citizens, businesses, and other arms of government. It aims to provide better service delivery, reduce operational costs, empower citizens, increase efficiency and transparency, and reduce corruption [8–11]. On the other hand, public administration deals primarily with the internal operations of the government agencies in charge of the administration of government operations [12]. Effective public administration involves managing digital technologies and online services to provide citizens with access to government information and services through electronic channels [13]. The digitization of public administration through the use of ICTS and the Internet to improve government service delivery is described as e-government [12,14]. E-government can enhance citizen participation, increase awareness of government initiatives, improve transparency, and reduce corruption [8,9,15].

The development of e-government is at varying stages worldwide. While most developed countries are comparatively advanced in the implementation and application of e-government for improving public-sector service delivery, most countries across the Africa Union member states are still confronted with the challenges of implementing e-government for public administration and governance. In the specific context of Ghana, the effectiveness of e-government in transforming public administration and governance requires critical evaluation. Issues such as limited access to technology, inadequate infrastructure, and the rising cost of data hinder e-government initiatives in Ghana. Although there are several studies on the performance of e-government in the European Union [16–18], there is a notable gap in the case of the African Union member states, specifically the impact of e-government on public administration and governance in the context of Ghana. While most advanced countries are comparatively advanced in their application of e-government for improving public-sector service delivery, most countries in Africa are still less developed in e-government development. To improve the development of e-government in these African countries, a comparative analysis is needed. Although there are several studies on the performance of e-government in the European Union [16–18], less is known for the case of African Union member states. Such a lack of clarity in the literature may hamper the performance of AU member states in delivering innovative public-sector services to citizens. To improve the development of e-government for public-service delivery in these African countries, a comparative analysis is needed, which, by extension, will contribute to the broader discourse on e-government in Africa and Ghana.

Hence, in this study, two main questions were raised: (1) What is the state of e-government development among the African Union member states? (2) What is the role of e-government as a catalyst for the transformation of effective public administration and governance in Ghana? These two main questions were analyzed and answered. First, the state of e-government development among African Union member states under the United Nations was examined, with the position of Ghana specifically determined. In this context, the Technique for Order Preference by Similarity to the Ideal Solution (TOPSIS), a method of multi-criteria evaluation of alternatives, was applied. The evaluation of e-government was based on the data generated by global institutions, including the United Nations, the International Telecommunication Union (ITU), and the World Bank. Second, the role of e-government as a catalyst for the transformation of effective public administration and governance in Ghana was assessed using interviews. This study thus contributes to the e-government literature, especially in understanding the nature of e-government development among African nations, with a special focus on its impact on the sustainability of public administration and governance systems in Ghana.

In the first section of the paper, a review of e-government policy and the various approaches to evaluating e-government is conducted. The second section is devoted to the presentation of the methodology applied in the study. The third section presents the results of both the quantitative and qualitative analyses. In the final section, discussions, conclusions, and policy implications are presented.

### 1.1. E-Government Policy Development in the African Union Member States

The African Union (AU), officially launched in 2002, is a continental body with membership from 55 African countries in the five regions of the continent. The AU recognizes the relevance of e-government as a crucial component of public administration and has adopted it as part of its information policy. E-government has since evolved over the years in all African countries from fragmented and inconsistent practices to more integrated and standardized methods [19].

Before the launch of AU, the African Information Society Initiative (AISI) was launched in 1996 with the aim to promote the growth and development of Africa's digital vision and improve living standards in areas such as education, healthcare, trade, job creation, and food security. Following the launch, by the year 2002/2003, the AU recognized the need for e-government to improve governance and service delivery and adopted a framework for e-government development [20,21]. The framework aimed to provide guidelines for developing e-government policies, strategies, and legal frameworks and establishing the necessary infrastructure and human capacity [20,21]. Subsequently, these African countries have consistently promoted e-government delivery in Africa. By 2007, the African Union E-Governance Program was launched to provide technical assistance to other member countries to develop and implement their e-government strategies. The program has tremendously facilitated the sharing of best practices and experiences among these member countries across the continent [22,23]. Moreover, the e-Africa Commission was also established to accelerate Africa's digital integration and create an information society and knowledge economy [24]. It was also established to provide equitable but affordable access to broadband and Internet services to promote the development of information and communication technology infrastructure in Africa [24,25].

Though progress was initially slow due to inadequate technological infrastructural investment, low-skilled labor, and a lack of political commitment to adopt and implement e-government practices [26,27], most African countries still learned lessons from Asia, Oceania, and Europe. These AU countries started implementing e-government policies as a way of delivering public services to their citizens and increasing accountability, transparency, and efficiency in their operations. In Asia, the United Arab Emirates, as the world's first 100% paperless government, has embraced digitalization, drastically reducing the time for family registration processing from 3 days to a few minutes and saving 10 million hours for business registration applicants [28,29]. For the case of Oceania, Fiji uses the digitalFIJI program launched the myFeedback platform for users to manage births, retrieve and apply for birth certificates, and register companies, providing an online space for discussions on governance and government services [29,30]. In Europe, Serbia's e-government implementation has significantly reduced the percentage of public-sector employees lacking basic computer skills to just 4%, with nearly all public institutions being equipped with data centers [17,29].

Additionally, though the dates of the establishment of government portals in each country vary, all aim to strengthen their commitment to e-government development [31]. The portals serve as central access points and aid in disseminating e-government information and services across member states. They further provide information on government services, programs, projects, and policies and also host online public services such as payment of taxes, passport applications, and business registration [20].

The development of e-government policies in Africa has had diverse impacts on digital transformation and public administration across Africa. Rwanda stands out with its Irembo platform as a success story, demonstrating the positive influence of robust e-government policies on public-service delivery and contributing to efficient public administration [32]. Kenyan citizens, through the creation of Huduma Centers across the country, are able to access various government services, digitally leading to a more efficient process, less bureaucracy, and higher levels of satisfaction among citizens [33]. South Africa's leveraging of e-government policies has used online platforms such as "eNatis" for vehicle registration and licensing and the "eFiling" system for tax submissions, therefore reducing bureaucratic

processes and enhancing accessibility to public services, ultimately improving the public administration system [34]. Similarly, the Ghana.gov server platform (https://www.ghana.gov.gh, Accessed on 1 September 2023) for public institutions has been able to generate results that meet the needs of citizens and enhance public administration.

Despite these efforts, the African Union continues to work with member countries to facilitate the development and implementation of effective digital transformation strategies that will enhance effective public administration and sustainable development on the African continent [35]. As technology continues to evolve, the AU aims to continue adapting and implementing e-government policies to meet the changing needs of citizens, businesses, and government agencies now and in the future.

*1.2. E-Government Policy Development in Ghana*

Ghana's position as one of the leaders in ICT development in sub-Saharan Africa is attributed to the unwavering dedication of successive governments' impactful initiatives and their implementation of policies to enhance digital infrastructure and promote access to government services through various online service platforms [4,36]. As a result, considerable efforts are consistently being made to ensure that growth in this sector is sustained [37]. Three of such projects are the e-Ghana Project, the e-Transform Project, and the Ghana Economic Transformation Project.

The first was the e-Ghana project, funded with USD 40 million from the World Bank in 2006; IT aimed to improve e-government interventions in Ghana through the use of ICT and public–private partnerships [38]. It had three components: creating an enabling environment, promoting IT investments and indigenous business development, and implementing e-government services and communication [38]. The project received further supplementary funding of USD 44.7 million in 2010 for another component. The project underwent a second restructuring in 2014 and ended on 30 December 2014 with a disbursement sum of USD 80.25 million from International Development Association resources [4,39]. The project aimed to improve Ghana's public-sector efficiency by implementing an integrated information system. The second project is the e-Transform Project, supported by the World Bank Group, as a government-led initiative aimed at enhancing economic development in Ghana by improving access to government services, increasing productivity in the agricultural sector, promoting SMEs, and improving the quality of education [40]. The third project is the Ghana Economic Transformation Project, aimed at enhancing a growth strategy that supports economic evolution by achieving increased investment rates and productivity growth across the economy, with a particular focus on non-resource-based sectors. Furthermore, it endeavors to generate employment opportunities and elevate incomes to ensure a superior quality of life [27]. Additionally, other initiatives including the Ghana Card Project, the launch of a portal known as the "Ghana.gov server" (https://www.ghana.gov.gh, Accessed on 1 September 2023), the National Digital Addressing System Project, etc., have facilitated gains for Ghana [21,41,42].

*1.3. Approaches to Monitoring E-Government Performance*

Several institutions and researchers are using varied approaches in the monitoring and evaluation of e-government performance across the various institutions by using different indicators at different times [16,43,44]. Some of the organizations involved in the Global South are the World Economic Forum, the International Telecommunication Union, the World Bank, USAID (Bringing Africa Online), and the United Nations (UN). The rankings have become unavoidable because they help track the progress and impact of electronic service delivery and improve transparency, accountability, and efficiency [45].

The World Economic Forum monitors and presents reports on the overall digital competitiveness of African countries using specific key indicators and promotes collaborative entrepreneurship to address global issues and shape agendas [46]. The ITU measures the digital development of countries based on skills, access, and use. It helps African countries to interconnect easily and expands access to ICTs for underserved areas [47]. E-government

rankings have gradually become very relevant because they have moved from merely measuring websites to critically evaluating applications and government qualities and guiding countries to focus on their efforts [48]. At the international level, Eurostat also analyzes and evaluates e-government data by using indicators that measure the relationships between citizens and businesses with public administration modernization. The project improves African integration through better statistics for informed decision making and fosters institutional capacity building [16,49].

The United Nations e-Governance Survey has become a constant biannual event among the 193 member countries across the globe. "The survey assesses global and regional e-government development through a comparative rating of national government portals relative to one another...it is designed to provide a snapshot of country trends and relative rankings of e-government development..." It helps African countries to learn from each other, identify areas of strength and challenges in e-government, and shape their policies and strategies [50].

From the literature above, all the organizations involved have different approaches to the monitoring of e-government performance. They all have different timeframes, methodologies, and monitoring indicators based on the purposes of their respective organizations. For the sake of consistency, we chose to focus on the approach used by the United Nations.

## 2. Materials and Methods

### 2.1. Study Approach, Data Source, and Analytical Tools

The analysis of e-government development is characterized by multivariate and complex methodologies, including ones that are both quantitative and descriptive [16,43,51,52]. A mixed-methods approach was therefore used in this study. The mixed-methods research design aims to obtain a deeper comprehension of phenomena by providing a complete picture [51]. The method is widely used because it combines both quantitative and qualitative data in a single study, providing a more robust inference than can be achieved using either of the approaches alone [51]. The analysis was therefore conducted using the mixed approach at two stages: (1) analyzing quantitative secondary data using multi-criteria decision making (MCDM) and (2) analyzing qualitative primary data from the interviews conducted.

First, quantitative secondary data were sourced from the UN e-Government Report [29] for sixteen (16) countries among the fifty-five (55) AU member states based on their e-government performance. The countries were chosen based on their performance indexes in online services, e-participation, human capital development, telecommunications infrastructure development, e-government development, and open government development. The selection also partly considered geographic and demographic diversity, economic status, and the population sizes of the countries with the aim of contributing to the generalizability and applicability of the study's findings to the entire AU region. The quantitative method based on positivism addressed the e-government development level and Ghana's ranking using multi-criteria decision-making (MCDM). The use of secondary data in research was well-grounded to guarantee reliable findings [53,54]. The method for solving multi-criteria decision making (MCDM) problems with TOPSIS was applied in this study [52]. The TOPSIS method was chosen over other methods because the TOPSIS method ensures that the ranking of variants is grounded on the relative similarity to the ideal solution [16,43]. TOPSIS considers the variability of the values involved and the distance from the ideal and basal variants. For ranking and selecting variants, TOPSIS has become a very essential and informative technique [55,56]. Due to its flexibility, potency, and efficacy as an application domain, it has become a well-known MCDM technique for solving both theoretical and real-life challenges [57]. Furthermore, it is widely applied because of its comprehensibility, simplicity, good computational efficiency, rationality, and ability to quantify relative performance using simple mathematical forms [58,59]. Despite our choice of TOPSIS, it is not without limitations. Specifically, it tends to overemphasize

certain attributes and sometimes lacks support for sensitivity analysis. Additionally, it may struggle when dealing with non-compensatory criteria [60].

Different software packages including Triptych package, SANNA-2014, and TOPSIS Solver are available for solving MCDM problems, and all require Microsoft Excel-2013 [61,62]. TOPSIS has been applied in supply chain management, manufacturing systems, marketing management, and other research areas [48]. For example, the TOPSIS method was applied in analyzing EU countries based on their economic and e-commerce performance [63]. The application of TOPSIS is founded on the concept that the selected alternative should have the shortest geometric distance from the best solution and the longest geometric distance from the worst solution. Applying the TOPSIS techniques, Ardielli and Halásková [16] and Ardielli [43] assessed e-government in EU countries, and the findings indicated that the level of e-government development has a positive linear correlation with good governance. Shown in Figure 1 is the study's model algorithm, detailing the steps involved in arriving at our final solution.

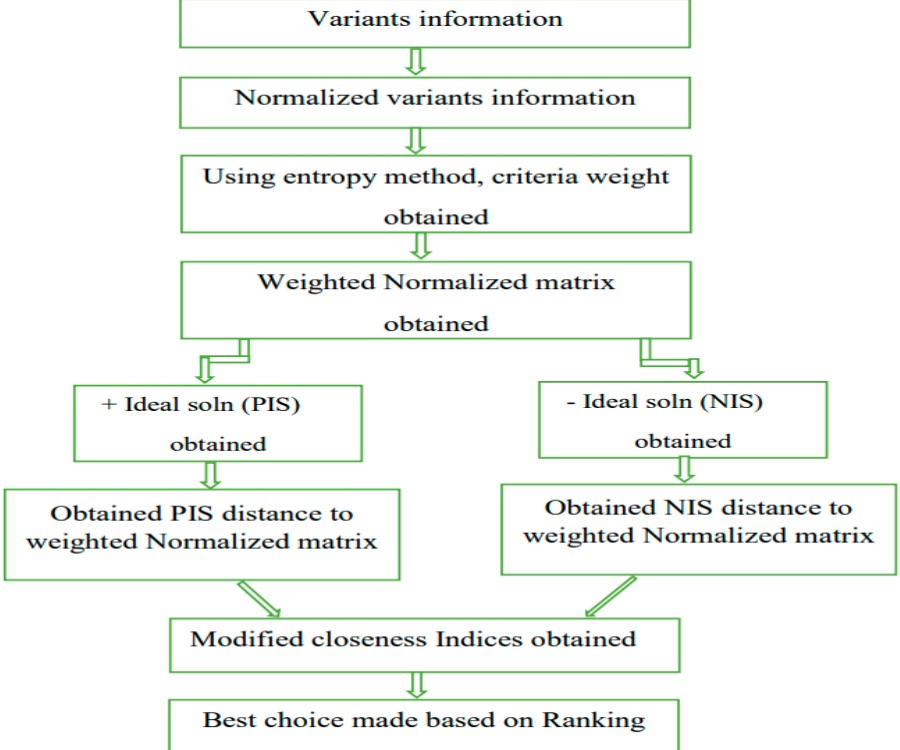

**Figure 1.** MCDM algorithm on e-government development in Africa.

### 2.2. Entropy and TOPSIS Model

Among the most popular models for MCDM evaluation is the entropy model. This model, aside from its direct role in MCDM evaluation, has been applied in many other areas such as, for example, fitting lost distribution functions in finance and insurance [64]. It can be applied to compute the weight of each item of data and estimate the amount of information each item contains. Thus, it is acknowledged that though a number of methods could have been applied, the entropy approach was adopted because of its robust nature in testing and ensuring a reliable output. This implies that the entropy weight, which is calculated from given raw data without the inclusion of subjective elements, reflects the advantages of entropy as an objective tool for attribute evaluation. The entropy model output is presented in the results section. The index weight offers advantages over alternative ways of determining index weight, is easily comprehensible and accepted by decision makers, and has high objectivity to assure the scientific nature of the evaluation closure [59,65].

In order to ascertain each attribute's influence on decision making, this study first computed its weight, and then, the weights of each criterion were compared. With $Q_1, Q_1, \ldots, Q_m$ representing the variant in this study, i.e., the selected AU member countries, and $H_1, H_1, \ldots, H_n$ representing decision criteria, a decision matrix was formed as shown below:

$$
\begin{array}{c}
\begin{array}{ccccccccc} H_1 & H_2 & . & . & . & . & . & H_j & H_n \end{array} \\
\begin{array}{c} Q_1 \\ Q_2 \\ . \\ . \\ Q_i \\ . \\ . \\ . \\ Q_m \end{array}
\left[
\begin{array}{ccccccccc}
H_{11} & H_{12} & . & . & . & . & . & H_{1j} & H_{1n} \\
H_{21} & H_{22} & . & . & . & . & . & H_{2j} & H_{2n} \\
. & . & . & & & . & . & . \\
. & . & . & . & . & . & . & . \\
. & . & & & & . & . & . \\
H_{i1} & H_{i2} & . & & . & H_{ij} & H_{in} \\
. & . & . & . & . & . & . \\
. & . & . & . & . & . & . & . \\
. & . & & & . & . & . \\
H_{m1} & H_{m2} & . & . & . & . & H_{mj} & H_{mn}
\end{array}
\right]
\end{array}
$$

After the decision matrix was formed, the next important step under entropy was to determine the target attribute denoted by $P_{ij}$, which was obtained as follows:

$$
P_{ij} = \frac{H_{ij}}{\sqrt{\sum\limits_{i=1}^{n} H_{ij}^2}} \tag{1}
$$

From (1), we defined $i = 1, \ldots, m; j = 1, \ldots, n$ as the number of variants and the decision criterion, respectively, and $H_{ij}$ as the value of the evaluation of variant $i$ with respect to decision criterion $j$. Again, the weight of each target attribute was needed and had to be determined. This is denoted by $V_j$, which ranges between 0 and 1 and is expressed as given:

$$
V_j = -k \sum\limits_{i=1}^{m} P_{ij} \ln P_{ij}, k = \frac{1}{\ln(m)} \tag{2}
$$

Finally, we determined the expected comparative equal weights and the attributes' targets levels as follows, respectively:

$$
w_{ij} = \frac{d_j}{\sum\limits_{j=1}^{n} d_j}, d_j = 1 - v_j \tag{3}
$$

On the other hand, as already mentioned, the TOPSIS approach to decision making is simple and intuitive, so decision makers can readily adopt it. One drawback of the strategy is that it requires combining it with other approaches in order to quantify the indications for situations that are not quantitative. As a result of evaluating solutions from both the positive and negative ideal views, the TOPSIS technique may prevent the drawback of neglecting needs from many angles and may guarantee the greatest possible solution. The weighted methodology was imperative in this study in obtaining the variants' objective weights for the criterion. Thus, through the integration of TOPSIS and entropy weighting, this study could objectively measure the outcomes and pinpoint high-weight values in order to determine the best solution. The TOSIS method by explanation was applied in the set of stages shown below:

First, vector normalization (Euclidean) was used for all criteria and thus for both benefits and costs, as defined below:

$$r_{ij} = \frac{x_{ij}}{\sqrt{\sum\limits_{i=1}^{m}(x_{ij})^2}} \tag{4}$$

where $r_{ij}$ represents the elements of the matrix, $R$; $i = 1, 2, \ldots m$; $j = 1, 2, \ldots r$; $x_{ij}$ is the original data input for variant $i$ and criterion $j$, and $m$ is the number of variants. Second, the weighted decision matrix $W$ was determined as follows:

$$w_{ij = V_j * r_{ij}} \tag{5}$$

Here, $w_{ij}$ denotes the weighted normalized value, and $V_j$ is the weight of the criterion obtained through the entropy method in (2).

Moving forward, the ideal variant $L_j$ and the baseline variant $D_j$ in relation with the matrix values $W$ were determined as below, where $J$ and $J'$ are, respectively, the benefit and cost indexes.

$$L_j = \{(\max w_{ij}/j \in J), (\min w_{ij}/j \in J'\epsilon\}$$
$$D_j = \{(\min w_{ij}/j \in J), (\max w_{ij}/j \in J'\epsilon\} \tag{6}$$

where $i = 1, 2, \ldots m$; $j = 1, 2, \ldots r$.

Consequently, the distance computations of variants from the ideal variant and baseline variant are, respectively, as follows:

$$d_i^+ = \sqrt{\sum\limits_{j=1}^{r}(w_{ij} - L_j)^2},$$
$$d_i^- = \sqrt{\sum\limits_{j=1}^{r}(w_{ij} - D_j)^2}, \tag{7}$$

Thus, for every $i = 1, 2, \ldots m$; $j = 1, 2, \ldots r$.

Finally, the relative distance indicator of variants from baseline variant was calculated as follows:

$$c_i = \frac{d_i^-}{d_i^- + d_i^+}, i = 1, 2, \ldots m. \tag{8}$$

Variants were arranged by non-growing values of $c_i$, and relying on the TOPSIS method, it was possible to rank the AU countries based on their performance of e-government and to validate the position of Ghana in the international comparison for the year 2022. From the study, the final lists of variants were the UN-16 countries, and the criteria were the six (6) e-government indicators ($i_1 - i_6$) selected [29]: (UN e-Government Report, 2022): (1) the online services index ($i$-$_1$), an indicator that tracks the progress of e-government services based on connectivity, availability, quality, and diversity of channels and the usage of these services by the citizens [66]; (2) the e-participatory index ($i$-$_2$), calculated as a supplementary index and used to broaden the dimension of the survey by concentrating on the use of online services to facilitate the provision of information by governments to citizens, engaging in decision making, and interacting with stakeholders; (3) the human capital index ($i$-$_3$), which estimates the influence of health and education on the productivity of the future generation of employees; (4) the telecommunications infrastructure index ($i$-$_4$), which is a composite indicator that assesses a country's readiness to capitalize on the opportunities offered by ICTs to enhance their competitiveness; (5) the e-government development index ($i$-$_5$), which is a composite indicator consisting of three indexes (the online service index, telecommunication index, and human capital index) that are equally weighted and cover a broad range of topics relevant to e-government; and (6) the open government development index ($i$-$_6$), which is computed as a supplementary index to

the online service index [29] and enhances the survey scope by focusing on the use of government data. OGDI is focused on policy, platform, and impact.

The study relies on information gathered from the [29]. The analyzed data describe the performance level of e-government in 2022. In the first stage, the data were inserted into the decision matrix $X$, where each element $xij$ demands the value of the $i$-th variant and of the $j$-th criteria [67], as shown in Table 1.

**Table 1.** Compiled data for 16 variants and 6 criteria—decision matrix X.

| Country | EGDI World Ranking | OSI ($i_1$) | EPI ($i_2$) | HCI ($i_3$) | TII ($i_4$) | EGDI ($i_5$) | OGDI ($i_6$) |
|---|---|---|---|---|---|---|---|
| Algeria | 112 | 0.3743 | 0.2273 | 0.6956 | 0.6133 | 0.5611 | 0.1972 |
| Botswana | 118 | 0.2740 | 0.1705 | 0.6932 | 0.6814 | 0.5495 | 0.2648 |
| Cabo Verde | 110 | 0.4965 | 0.25 | 0.6507 | 0.5507 | 0.5660 | 0.3803 |
| Côte d'Ivoire | 120 | 0.5467 | 0.375 | 0.5748 | 0.5186 | 0.5467 | 0.7606 |
| Egypt | 103 | 0.5730 | 0.3523 | 0.6375 | 0.5579 | 0.5895 | 0.3127 |
| Gabon | 116 | 0.3578 | 0.2045 | 0.6706 | 0.6279 | 0.5521 | 0.2141 |
| Ghana | 106 | 0.5361 | 0.4545 | 0.6176 | 0.5934 | 0.5824 | 0.8310 |
| Kenya | 113 | 0.6821 | 0.5795 | 0.5641 | 0.4305 | 0.5589 | 0.7268 |
| Mauritius | 75 | 0.6282 | 0.4205 | 0.7733 | 0.7588 | 0.7201 | 0.7296 |
| Morocco | 101 | 0.4721 | 0.2727 | 0.6350 | 0.6676 | 0.5915 | 0.4366 |
| Namibia | 121 | 0.4316 | 0.25 | 0.6516 | 0.5133 | 0.5322 | 0.1775 |
| Rwanda | 119 | 0.7935 | 0.6364 | 0.5322 | 0.3209 | 0.5489 | 0.7070 |
| Seychelles | 85 | 0.4424 | 0.2273 | 0.7758 | 0.8198 | 0.6793 | 0.3296 |
| South Africa | 65 | 0.7487 | 0.5909 | 0.7733 | 0.6850 | 0.7357 | 0.7662 |
| Tunisia | 88 | 0.6031 | 0.5455 | 0.6911 | 0.6646 | 0.6530 | 0.7606 |
| Zambia | 131 | 0.4414 | 0.375 | 0.6744 | 0.3909 | 0.5022 | 0.5493 |

Every criterion in TOPSIS is maximized by default, implying that the greater the value, the more improved its rating. In computing TOPSIS, the weight of each criterion is taken into account. Using the scoring method, the results had the same weights and were appropriate for the assessment of AU member states. SANNA, an MS Excel-based add-in application, was used for the computation of the results [61].

Secondly, the study also performed an empirical study based on semi-structured one-on-one interviews with thirty (30) individuals and decision makers in the private and public sectors, academia, and civil society. The interviewees were participants attending the Ghana Internet Governance Forum, an annual event that brings together stakeholders with a common interest in e-government and Internet governance. Using purposive sampling methodology, the respondents were selected from each of the organizations represented at the forum based on each respondent's expertise in Internet governance and public administration. The semi-structured interviews with a qualitative nature allowed the participants to informally share in-depth knowledge, experiences, and perspectives [68]. The qualitative method was underpinned by interpretivism and interviews with individuals and/or focus groups that addressed the impact of e-government on public administration and governance [69]. The qualitative research approach described in-depth the experiences of respondents and interpreted the significance of their actions and feelings [70].

Each interview commenced with a brief description of e-government and effective public administration and governance as discussed in the literature. Beyond the definitions, we then asked the respondent the extent to which each of the various good governance indicators (effectiveness and efficiency, responsiveness, information dissemination, cost-effectiveness, transparency, accountability, flexibility, and security) from Rothstein and Teorell [71] and the United Nations [72] impact effective public administration, juxtaposing it with the pre-e-government (manual system of administration) in Ghana, and how e-government service changes the provision of public-service delivery as compared to the pre-e-government era. The interview with each respondent was recorded and lasted about 10 min.

## 3. Results

This section is divided into two: First, results from the comparison of the level of e-government development in the year 2022 as a benchmark are presented using a multi-criteria analysis method. The analysis was conducted for 16 AU member countries in the UN e-Government Development Report and the results presented. The findings showed the state of e-government in individual African countries in comparison with the United Nations averages and the ranking of Ghana. Secondly, using data from the conducted interviews, the results of the impact of e-government on Ghana's public administration are also presented.

As presented in the review of the methodology, the state of e-government can be considered a multivariate phenomenon. Hence, for the evaluation, six (6) variables, namely $i$-1 to $i$-6, were applied. The evaluation variables with their classification and descriptive statistics are also presented. The distance coefficient of variant $i$ from the ideal variant $d_i^+$, the distance coefficient of variant $i$ from the basal variant $d_i^-$, the relative distance indicator $c_i$, and the relative distance of variant $i$ from the basal variant were all calculated [73,74]. The summarized results are presented below.

Table 2 shows the reliable output of the entropy model. From the results the entropy weight, which is calculated from the raw data without the inclusion of subjective elements, reflects the advantages of entropy as an objective tool for attribute evaluation.

**Table 2.** Statistics of the entropy model.

|  | OSI(C-1) | EPI(C-2) | HCI(C-3) | TII(C-4) | EGDI(C-5) | OGDI(C-6) |
|---|---|---|---|---|---|---|
| Vj | 0.3348 | 0.3207 | 0.3446 | 0.3382 | 0.3445 | 0.3109 |
| dj | 0.6652 | 0.6793 | 0.6554 | 0.6618 | 0.6555 | 0.6891 |
| wj | 0.1660 | 0.1696 | 0.1636 | 0.1652 | 0.1636 | 0.1720 |

From the results in Table 3, it can be seen that the values of the distance coefficients and the relative distance indicators characterize the distance between the two countries based on their e-government development. The evaluated indicator $C_i$ values ranged from 0 to 1, where value 0 represents the basal variant, whereas value 1 represents the ideal variant [43,75]. The results show the performance e-government development of each country using the relative distance indicator, with ranking as follows: Botswana, Gabon, Namibia, Algeria, Cabo Verde, Seychelles, Zambia, Tunisia, Morocco, Egypt, Côte d'Ivoire, Rwanda, Kenya, Ghana, Mauritius, and South Africa. This implies that there is some level of connectedness in their use of e-government or ICTs for public-service delivery [76]. In the findings of Dias [77], it was recommended that low-income countries should build sustainable cooperation with experienced foreign partners in the implementation of e-government strategies and plans to ensure effective public administration and governance.

**Table 3.** Distance coefficients values and relative distance indicator.

| Country | $d_i^+$ | $d_i^-$ | $C_i$ | Country | $d_i^+$ | $d_i^-$ | $C_i$ |
|---|---|---|---|---|---|---|---|
| South Africa | 0.0120 | 0.0756 | 0.8635 | Algeria | 0.0687 | 0.0260 | 0.2744 |
| Mauritius | 0.0258 | 0.0643 | 0.7136 | Kenya | 0.0363 | 0.0626 | 0.6331 |
| Seychelles | 0.0581 | 0.0451 | 0.4369 | Gabon | 0.0701 | 0.0254 | 0.2664 |
| Tunisia | 0.0538 | 0.0651 | 0.5477 | Botswana | 0.0730 | 0.0289 | 0.2834 |
| Morocco | 0.0533 | 0.0365 | 0.4063 | Rwanda | 0.0426 | 0.0696 | 0.6202 |
| Egypt | 0.0522 | 0.0362 | 0.4096 | Côte d'Ivoire | 0.0434 | 0.0497 | 0.5339 |
| Ghana | 0.0350 | 0.0583 | 0.6248 | Namibia | 0.0689 | 0.0217 | 0.2394 |
| Cabo Verde | 0.0582 | 0.0298 | 0.3390 | Zambia | 0.0550 | 0.0354 | 0.3913 |

The results also show the development of e-government in the five regions of Africa using the highest cumulated coefficient values. Among the regions, Northern Africa (Tunisia, Morocco, Egypt, and Algeria) has the highest cumulated coefficient values. Eastern Africa

(Mauritius, Seychelles, Kenya, and Rwanda) follows with a very close distance coefficient among the countries. Southern Africa (South Africa, Botswana, Namibia, and Zambia) is next, with a wide gap distance coefficient between South Africa and Namibia. Western Africa (Ghana, Cabo Verde, and Cote d'Ivoire) follows, with a wide gap between Ghana and Cote d'Ivoire. Central Africa (Gabon) has only one country with an average distance coefficient compared to other countries. This implies that the level of investment in ICT and e-government infrastructure determines its impact on public administration and governance. This was consistent with the findings that if developing countries systematically and successfully integrate ICTs into governance structures, public administration will evolve and impact government operations and citizen engagement [78].

Further employing the TOPSIS method and using the distance coefficient ($Ci$) values, the countries were further ranked into three categories by applying Ward's method of hierarchical clustering [43,79]. According to the results shown in Figure 2 and Table 4, the countries were classified into three categories based on their e-government performance. The three categories in Table 4 include the above-average-, average-, and below-average-performing countries. The above-average position is occupied by South Africa, Mauritius, the Seychelles, Tunisia, Morocco, Egypt, and Ghana. The below-average countries include Cape Verde, Algeria, Kenya, Gabon, Botswana, Rwanda, Cote d'Ivoire, Namibia, and Zambia. Among the Western African countries, Ghana can be said to be the best-performing country. The result of the analysis was consistent with other findings by TOPSIS using the Ward method. Generally, all countries in the clusters were closely ranked. Through different variables that were, however, similar in terms of analysis, Vasylieva et al. [80] found that countries with similar socioeconomic trends show similar relationships between environmental tax revenues and GDP growth rates, indicating a need for tailored tax system designs.

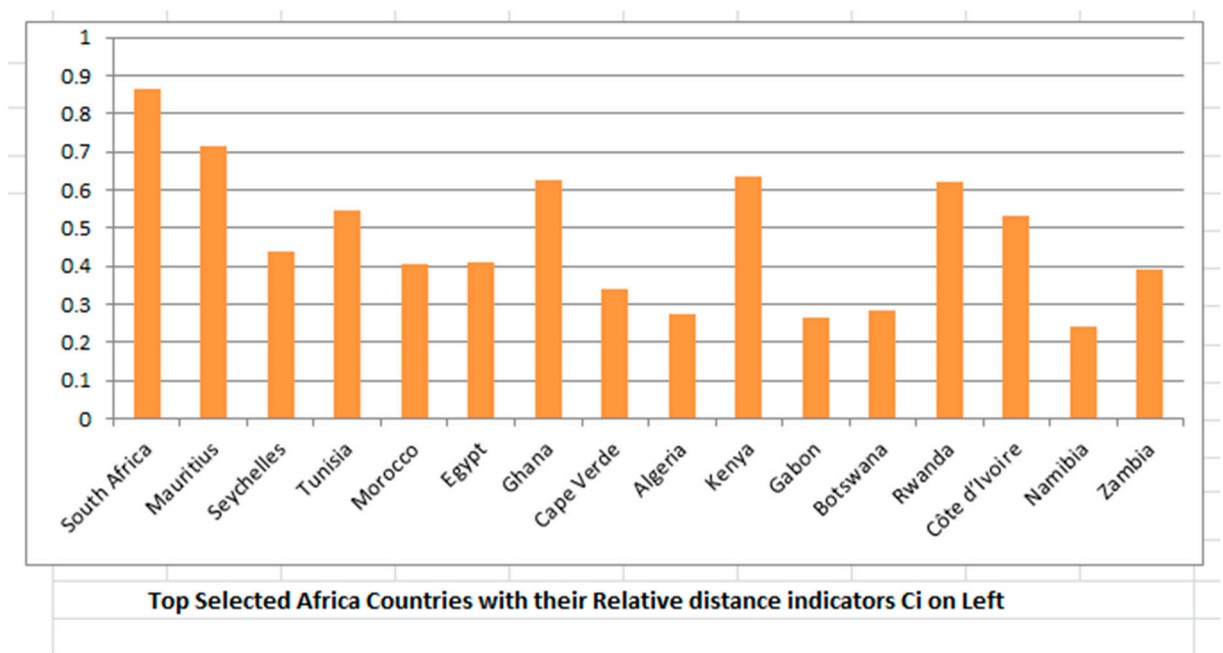

**Figure 2.** Dendrogram presenting the evaluation of e-government in the selected countries.

**Table 4.** Grouping of United Nations Member States per e-Government Development.

| Above-Average-Performing Countries | Average-Performing Countries | Below-Average-Performing Countries |
|---|---|---|
| South Africa, Mauritius, Ghana, Kenya, and Rwanda | Seychelles, Cote d'Ivoire, Tunisia, Morocco, Egypt, Cabo Verde, and Zambia | Algeria, Gabon, Botswana, And Namibia |

These values can be useful for policymakers to determine the extent of economic cooperation that can be established between different countries.

Secondly, the respondents provided the study with an extensive database from the recorded interviews. The recordings were carefully decoded, and the most informative content with relevance to the research was extracted and transcribed under the various indicators contained in Table 5. The analysis was further extended to the participant's responses on the impact of e-government on public administration and governance in comparison with pre-e-government (manual system of administration) using indicators from the general good governance principles [71,72]. The respondents were resolute that e-government is a critical enabler for effective public administration and governance in Ghana. E-government improves public administration quality in Ghana by reducing costs, expanding access, increasing efficiency, and achieving customer satisfaction [21]. Table 5 is a summary of the various responses grouped into themes regarding how e-government contributes to Ghana's effective public administration and governance.

**Table 5.** Impact of e-Government Services on Public Administration and Governance.

| Indicators | e-Government Impact | Manual System Impact |
|---|---|---|
| Effectiveness and efficiency | Faster and more efficient due to automation Provides 24/7 access to services and information Ghana.gov server platform (https://www.ghana.gov.gh accessed on 1 September 2023) | Slower due to manual processes and paperwork Limited access to services and information due to office hours and physical location |
| Responsiveness | Swift response to public-service users' complaints | Delay in response to complaints |
| Information dissemination | Efficient and effective dissemination of information | Ineffective information dissemination |
| Cost-effectiveness | Reduced cost by reducing paperwork and staff needed | More expensive due to higher staffing and paperwork cost |
| Transparency | Online access to information on the implementation of policies, decisions, and results of elections | Lacks transparency due to manual processes and limited public access to information |
| Accountability | Can improve accountability through tracking and audit trails | Limited accountability due to manual processes and bureaucratic structures |
| Flexibility | Can be more flexible and adaptable to changing needs | Limited flexibility due to manual processes and bureaucratic structures |
| Security | Provides robust security measures to protect sensitive information | Vulnerable to data breaches and security threats, theft, fraud, and limited security measures |

Source: Participants' responses.

## 4. Discussion

The current study, using the TOPSIS multicriteria evaluation technique, analyzed the level of e-government development among the African Union member countries and determined the position of Ghana. It further examined how the state of Ghana's e-government development affects the transformation of public administration delivery and governance.

The findings from the relative distance indicators and distance coefficients using the MCDM by TOPSIS shown in Tables 3 and 4, and Figure 2 are consistent with the findings of other scholars and major international organizations, such as Ardielli and Halásková [16], ITU, 2020, the World Economic Forum [46], the World Bank [81], and the Mo Ibrahim Report [82]. Though there are similarities and differences in the levels of e-government development across African countries, such as variations in the number of Internet users, gender gap in favor of men, Internet penetration, and access to Internet broadband, most of the countries lack proper governance for innovation and, in particular, ICT [83,84].

Africa made the most significant e-government development progress in the 2022 report. Africa's EGDI value increased by 3.6% as compared to the other continents of

Europe (1.7%), the Americas (1.5%), and Asia (1.9%). Additionally, Africa's TII also rose by 12% compared to the Americas (6.5%) and Asia (4.6%); this may have contributed to the higher EGDI averages in these regions. Similarly, Africa also experienced a high variance phenomenon. Out of the 54 countries, only 4 had an EGDP value above the global average. The rest of the countries had lower values, indicating the e-government development gaps and the continued existence of the digital divide in the continent. These patterns are consistent with the previous surveys in 2020 and 2018. Regarding EGDP groupings, while there is no African country in the very-high-EGDP group, 30% are in the high-EGDP category and 59% in the middle group.

Furthermore, the results presented on the impacts of e-government on public administration were consistent with finding of Mensah [12] and Brown and Toze [85], including citizen-centered services, new skills and relationships, information as a public resource, and impact on accountability and management models.

Regarding effectiveness and efficiency, the respondents were emphatic that e-government has made it more feasible through the Ghana.gov server platform (https://www.ghana.gov.gh, accessed on 1 September 2023) for a public institution to generate timely results that meet the needs of citizens than through the manual system, which was characterized by a lot of paperwork. Through the portal, public services are available to citizens 24/7. The efficiency in the delivery of business registration, renewal of driving licenses, payment of taxes, clearing of goods at the port, and public procurement was emphasized. Citizen-centered service delivery is a critical concept in ensuring that public services are designed to meet citizens' needs and encourage them to participate in civic duties [86]. The client-centered model includes self-service as an important component that allows for 24/7 service provision. Business processes and reengineering concepts are used in this transformation of service delivery [86].

Regarding responsiveness and information dissemination, e-government ensures that requests for information on public services and complaints by the public are swiftly answered within a reasonable period. Respondents cited moments in which citizen complaints received via online platforms including service portals, emails, and phone calls were swiftly handled. The public complaint service application known as the Spring Boot microservice architecture offers a flexible and scalable solution for e-government services, making it easy to swiftly respond to public inquiries [87]. Information as a public resource strengthens public-sector administration systems to manage large amounts of information and data, which are considered key resources for enacting effective legislation, policies, and institutions [12]. It has significantly transformed government information management and documentation through digitization, ensuring proper collection, production, storage, retrieval, dissemination, protection, and disposal of government information [85].

Concerning participation, e-government ensures that the voices of the vulnerable in society are considered in public decision making. Individuals with the right skills are permitted to contribute to issues in public administration. Some of the online participation platforms include social media, emails, WhatsApp group pages, etc. E-government requires innovative skills and planned relationships for both public-sector employees and the government as a whole. E-government allows for collaborative working techniques and information sharing in public administration, requiring public-sector workers to adapt and develop new skills. The government also builds strategic relationships with the private sector to develop e-government infrastructure, seek advice, and tap into private-sector skills and capabilities to deliver public services more effectively [86,88]. Some challenges in the Ghanaian public sector drive the need for new working relationships with private-sector organizations to be resolved.

Regarding transparency and accountability, respondents believed that e-government provides information on government implementation of policies, decision making, and presentation of election results to the public in such a way as to enable it to effectively follow and contribute to the work of the local authority. Through government departments such as State Interests and Governance Authority (SIGA) and the Ghana Audit Service (GAS),

transparency and accountability are assured. SIGA provides information on the efficiency and profitability of state-owned enterprises engaged in some form of trade and industry, and the GAS provides audited report reports of all government institutions annually. This finding was backed by [86], who found that e-government enhances the accountability of government and other sector agencies by deepening and changing the working relationship among the public and civil servants and the representatives elected [86]. E-government is a solution to the problem of inadequate accountability in public administration. It promotes transparency measures, financial accountability, and openness to increase citizen participation in governance [89]. Through easy access to information and data such as budgets and expenditure statements, e-government improves transparency in decision-making processes. The online services and application-tracking facilitate administrative tasks such as treasury management and human resource management, ultimately enhancing efficiency in public administration. Adam [90] found that though e-government development in Africa minimizes corruption and improves the quality of public administration, its direct effects on corruption were not significant.

*Policy Implication*

As a result of inadequate infrastructure, some countries were found to have not performed well in their e-government development.

The Ministry of Communications and Digitalization should therefore advocate for increased investment in the development and maintenance of robust e-government infrastructure. The National Information Technology Agency (NITA) should ensure reliable Internet connectivity, interoperability among various government systems, and the upgrading of technological capabilities.

Comprehensive training programs for staff in the Ghana Revenue Authority, Driver Vehicle Licensing Authority, and the Registrar General Department, when encouraged, would enhance their digital competencies. When empowered, these public officials can effectively manage the e-government systems to ensure efficient public-service delivery.

To ensure the enhanced impact of e-government on public administration, NITA should partner with the Civil Society Organization to implement robust monitoring and evaluation mechanisms for regularly assessing the effectiveness of e-government initiatives. Key performance indicators of good governance practices should be aligned to e-government objectives and regularly be reviewed to make informed policy adjustments. Public awareness campaigns could be launched to educate citizens on the benefits of using e-government services. This can enhance the adoption of and trust in the use of e-government platforms.

These policy contributions, when implemented, can contribute to the successful integration of e-government in Ghana by fostering and improving public administration and governance across Ghana and other regions as well.

## 5. Conclusions

This study investigated the development of e-government among the African Union member states and determined the ranking of Ghana. Overall, the evaluation showed that the state of e-government in the African Union is currently more advanced than found in previous reports, with significant differences between countries, and there is room for improvement in many areas. The evaluation also highlights the importance of using multicriteria evaluation methods to assess the performance of e-government systems, as this can provide a more comprehensive and nuanced picture of their strengths and weaknesses.

This study underscores the positive impact of e-government development on increasing the efficiency and effectiveness, transparency, accountability, participation, and responsiveness of public administration. E-government platforms have streamlined processes, reduced bureaucratic bottlenecks, and enhanced the accessibility of public services. This is evident through the transformation of service provision at the various ministries, departments, and agencies in the public sector. Through the integration of technology,

Ghana provides a digital services and payments platform called the Ghana.gov server (https://www.ghana.gov.gh, accessed on 1 September 2023) for most public services. Ghana has also aligned its long-term digital policies to its national goals. This has transformed the traditional concept of governance toward electronic means. More ministries, departments, and agencies are still being transformed.

For the sustainability of the transformation, there is a need to foster interagency collaboration to create a seamless and integrated governance system among major stakeholders in both the public and private sectors. Breaking down silos and promoting data-sharing mechanisms can lead to more efficient and responsive public administration. The study points to both the progress and challenges of e-government. Actionable insights are therefore offered for policymakers, emphasizing the relevance of addressing infrastructure gaps, improving monitoring and evaluation, promoting digital inclusion, and adopting user-centric approaches for a sustained positive impact on public administration and governance in Ghana and the wider region.

The study is not without limitation, as sixteen (16) countries were selected from the five regions of the African Union member states for the study. However, to fully comprehend the state of e-government in Africa, future studies should consider more African countries in their study and also adopt different approaches in their analysis of the data.

**Author Contributions:** B.J.T., conceptualization, analysis, and drafting; Z.T., supervision, review and editing, and funding acquisition; J.A., data analysis and review; J.C.D., review and editing; S.N.-A.O., data processing and review. All authors have read and agreed to the published version of the manuscript.

**Funding:** This research received no external funding.

**Institutional Review Board Statement:** Not applicable.

**Informed Consent Statement:** Not applicable.

**Data Availability Statement:** Primary data is contained within the article, and secondary data is available upon request.

**Conflicts of Interest:** The authors declare no conflicts of interest.

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
