# Peer review of "Evaluating E-Government Development among Africa Union Member States: An Analysis of the Impact of E-Government on Public Administration and Governance in Ghana"

_sustainability, doi:10.3390/su16031333_

Round 1

Reviewer 1 Report

Comments and Suggestions for Authors

Thank you for inviting me as a reviewer for the paper. The manuscript is contents fit with the journal’s topics. In this manuscript authors examine e-government development across African Union member states, focusing on the case of Ghana, and the role of e-government in the reform of public administration and governance in Ghana. Using the MCDM model entropy-TOPSIS, the study analyzes the key e-government indicator indexes.  The idea of the paper is interesting, but there are a number of omissions in the manuscript:

 . The abstract completely needs to be rewritten. The abstract should include the article's main (1) impact and (2) significance on decision-making systems. Note that a good abstract should contain aims, methods, findings, and recommendations.  In addition, it should cover five main elements, introduction, problem statement, methodology, contributions, and results.

. The literature analysis needs to be improved. The analysis relies on works from an older date (2020 and older), only a few works from the period 2022-2023. It is necessary to analyze in detail 10-20 recent papers (2022-2023). Based on literature analysis the authors need to discuss their contributions compared to those in related papers. The research gap and motivation should be clarified in the introduction section. Authors should begin with the problem, the gap, then propose the research question and just after that say what they want to do to address that. Where is the gap? And you should clearly why it is a gap?

. At the end of the introduction, announce the rest sections of the paper in one paragraph.

. A model flowchart is missing.

. The description of the applied methods in the model is incomplete. The authors only mention the Entropy method in one place (on the basis of which they calculate the weighting coefficients). A separate unit should be separated for this method, as well as for the TOPSIS method. Also, both methods should be shown through steps.

. The authors state that Table 1 was created as a result of the Author's estimation. A serious study requires research, and the least that should have been done was gathering the opinions of experts.

. All results, including table 1, should be presented in Section 3 (Results). The description of criteria (indicators) should also be in section 3.

. A sensitivity analysis is missing. A sensitivity analysis should be done using the change in weighting coefficients. Considering that the Entropy method was used to define the weighting coefficients, the sensitivity analysis is very important in the research.

. What about comparing the results with other methods?

. What are the limitations of the model?

. No directions for future research are given.

Author Response

Thank you for the valuable inputs

Reviewer #1:

Comments and Suggestions for Authors

  1. The abstract completely needs to be rewritten. The abstract should include the article's main (1) impact and (2) significance on decision-making systems.

Note that a good abstract should contain aims, methods, findings, and recommendations.  

In addition, it should cover five main elements, introduction, problem statement, methodology, contributions, and results.

The abstract has been revised to capture aims, methods, findings, and recommendations.  

  1. The literature analysis needs to be improved. The analysis relies on works from an older date (2020 and older), only a few works from the period 2022-2023. It is necessary to analyze in detail 10-20 recent papers (2022-2023).

Thank you, current references for the period 2022-2023 have been revised to more than 10 references 

Based on literature analysis the authors need to discuss their contributions compared to those in related papers. The research gap and motivation should be clarified in the introduction section. Authors should begin with the problem, the gap, then propose the research question and just after that say what they want to do to address that. Where is the gap? And you should clearly why it is a gap?

The research gap, motivation, problem, and research questions revised. (pages 2 & 3, paragraphs 4 & 5).

  1. At the end of the introduction, announce the rest sections of the paper in one paragraph.

The rest of the sections of the paper have been presented in the last paragraph of the introduction. (page 3, last paragraph)

  1. A model flowchart is missing.

Based on your suggestion, we have included all the steps involved in our adopted models under Materials and Methods. (pages 7&8)

  1. The description of the applied methods in the model is incomplete. The authors only mention the Entropy method in one place (on the basis of which they calculate the weighting coefficients). A separate unit should be separated for this method, as well as for the TOPSIS method. Also, both methods should be shown through steps.

We have in the revision, separated the entropy from the TOPSIS as suggested. The requested steps for these two models are also shown in this modified work. (page 6&7). Additionally, the entropy’s model statistics are also included. (page 10, Table 2)

  1. The authors state that Table 1 was created as a result of the Author's estimation. A serious study requires research, and the least that should have been done was gathering the opinions of experts.

The inscription beneath Table 1 as observed was an oversight and we tremendously thank you for drawing our attention. Table 1 is the raw data for the 16 variants and 6 criteria. Since the data has been sourced from the e-government report, 2022, and the interpretation done based on research from other literature, the “Author's estimation” has been deleted leaving the title of the table. (page 9, Table 1)

  1. All results, including Table 1, should be presented in Section 3 (Results). The description of criteria (indicators) should also be in section 3.

With your advice, we have ensured that all results and related materials are presented in section 3. However, table 1 is sourced data for the calculations leading to the results presented. Hence Table 1 has not been added to the Result Section 3. (page 9)

  1. A sensitivity analysis is missing. A sensitivity analysis should be done using the change in weighting coefficients. Considering that the Entropy method was used to define the weighting coefficients, the sensitivity analysis is very important in the research.

We appreciate your expert’s suggestion. Following your earlier advice on the entropy model, we have captured in this modification the significant role of the entropy model.

It can be applied to compute the weight of each item of data and estimate the amount of information it contains. Thus, it is acknowledged that though the SENSITIVITY procedure could have been applied, the entropy approach was adopted because of its robust nature in testing and ensuring a reliable output. Implying that the Entropy weight, which is calculated from a given raw data without the inclusion of subjective elements, reflects the advantages of Entropy as an objective tool for attribute evaluation. The index weight offers advantages over alternative ways of determining index weight, is easily comprehensible and accepted by decision-makers, and has high objectivity to assure the scientific nature of the evaluation closure. Thus, through the integration of TOPSIS and entropy weighting, this paper can objectively measure the outcomes and pinpoint high-weight values in order to determine the best solution.

  1. What about comparing the results with other methods?

An additional comparison of the results has been done (pages 11 & 12).

  1. What are the limitations of the model?

Limitations of the model have been revised (page 6, paragraph 2).

  1. No directions for future research are given.

Directions for future research revised (page 16, last paragraph)

Reviewer 2 Report

Comments and Suggestions for Authors

Dear authors, Thank you for the interesting material.

Let me make comments that may improve the article.

This article is written in good scientific language, it is well structured, which allows the reader to easily familiarize themselves with the material and understand both the intention of the authors and the goals of the article, research objectives and research methodology.

HOWEVER, there are some wishes.

First, it might be useful to indicate whether there is experience elsewhere in the world in implementing e-government that African countries could learn from.

Secondly, at the end of the work, further prospects for research and limitations of this study are usually indicated.

I wish you successful scientific publications

Author Response

Reviewer #2:

Comments and Suggestions for Authors

  1. First, it might be useful to indicate whether there is experience elsewhere in the world in implementing e-government that African countries could learn from.

E-government implementation experiences elsewhere in the world have been revised (page 3, under 1.1 last paragraph).

  1. Secondly, at the end of the work, further prospects for research and limitations of this study are usually indicated.

Further prospects for research and limitations have been revised (page 16, last paragraph)

Reviewer 3 Report

Comments and Suggestions for Authors

Make reference to the influence of the African Information Society Iniciative (AISA) on Ghana's e-government.

Identification, among the 30 individuals and decision-makers, those who are directly related to the transformation of digital governance in Ghana - identification of transformation clicks, Relationship between interviewees and digital government transformers; identification of constraints, barriers, blockages to digital transformation and identification of e-governance enablers.

1. What is the main question addressed by the research?

The level of development of e-government in Ghana, in the context of African Union countries, rather than the impact of digital government in public e-service quality.

2. Do you consider the topic original or relevant in the area?

Yes.

Does this address a specific gap in the field?

The authors could go further if they assessed the impacts of digital government in Ghana.

3. What does it add to the subject area compared to other publications material?

A case study.

4. What specific improvements should authors consider in relation to the methodology?

Introducing an explanation about the method of selecting interviewees and the semi-structured survey guide is superficial, and explaining in the introduction all research methods and techniques

Author Response

Reviewer #3:

Comments and Suggestions for Authors

Make reference to the influence of the African Information Society Iniciative (AISA) on Ghana's e-government.

The influence of AISA on Africa's e-government development has been revised (page 3, under 1.1 second paragraph

Identification, among the 30 individuals and decision-makers, those who are directly related to the transformation of digital governance in Ghana - identification of transformation clicks, Relationship between interviewees and digital government transformers; identification of constraints, barriers, blockages to digital transformation and identification of e-governance enablers.

  1. What is the main question addressed by the research?

The level of development of e-government in Ghana, in the context of African Union countries, rather than the impact of digital government in public e-service quality.

  1. Do you consider the topic original or relevant in the area?

Yes.

Does this address a specific gap in the field?

The authors could go further if they assessed the impacts of digital government in Ghana.

Discussion on the impacts of digital government in Ghana has been expanded (pages 14 & 15).

  1. What does it add to the subject area compared to other publications material?

A case study.

  1. What specific improvements should authors consider in relation to the methodology?

 Introducing an explanation about the method of selecting interviewees and the semi-structured survey guide is superficial, and explaining in the introduction all research methods and techniques.

 The explanation of the method of selecting the interviewees has been expanded (page 10, first paragraph).

The research methods and techniques have been mentioned briefly while a detailed explanation is given under the Materials and Methods (pages 5-10).

Reviewer 4 Report

Comments and Suggestions for Authors

Thanking the authors for their contribution, I have the following concerns to be addressed.

1. Abstract: The abstract provides a concise overview but could be improved by offering more specific insights into the study's unique contributions and findings, particularly regarding the case of Ghana.

2. Introduction: The introduction effectively sets the context but could be enhanced by more directly linking global trends in e-government development to the specific focus of the study on African Union member states and Ghana.

3. E-Government Policy Development in the African Union Member States: This section could benefit from a more detailed analysis of how these policies have directly impacted e-government implementation and public administration in various member states.

4. E-Government Policy Development in Ghana: The discussion on Ghana's policy development is informative. Further elaboration on how these policies compare with other African Union states in terms of effectiveness could add depth to the analysis.

5. Approaches to Monitoring E-Government Performance: While the overview of different monitoring approaches is comprehensive, integrating a critical analysis of these methods' effectiveness in the African context would enhance this section.

6. Materials and Methods: The methodology is well-explained but could be improved by providing more justification for the choice of countries included in the study and a deeper explanation of the analytical tools used, particularly the TOPSIS method.

7. Results: The results are clearly presented, but further discussion on how these findings relate to broader trends in e-government development in Africa would be beneficial.

8. Discussion: The discussion makes good connections with existing literature. It could be strengthened by more explicitly linking the results to the implications for future e-government strategies in Ghana and other African Union member states.

Comments on the Quality of English Language

Overall the English language used is good but it could be improved to improve its readability. 

Author Response

Reviewer #4:

Thank you for your valuable comments 

Comments and Suggestions for Authors

Thanking the authors for their contribution, I have the following concerns to be addressed.

1- Abstract: The abstract provides a concise overview but could be improved by offering more specific insights into the study's unique contributions and findings, particularly regarding the case of Ghana.

The abstract has been revised to capture the contributions and findings

  1. Introduction: The introduction effectively sets the context but could be enhanced by more directly linking global trends in e-government development to the specific focus of the study on African Union member states and Ghana.

The INTRODUCTION has been revised to capture lessons learned across the globe and narrowed to the African Union member states and Ghana (page 3, paragraphs 2&5).

  1. E-Government Policy Development in the African Union Member States: This section could benefit from a more detailed analysis of how these policies have directly impacted e-government implementation and public administration in various member states.

The impact of the policies on e-government implementation and public administration in the selected member states has been addressed (page 4, paragraph 3).

  1. E-Government Policy Development in Ghana: The discussion on Ghana's policy development is informative. Further elaboration on how these policies compare with other African Union states in terms of effectiveness could add depth to the analysis.

Further elaboration on the impact of the policies compared with other African Union States has been revised (page 4, paragraphs).

  1. Approaches to Monitoring E-Government Performance: While the overview of different monitoring approaches is comprehensive, integrating a critical analysis of these methods' effectiveness in the African context would enhance this section.

An additional explanation of the effectiveness of the methods in the African context has been expanded (page 6, under 1.3).

  1. Materials and Methods: The methodology is well-explained but could be improved by providing more justification for the choice of countries included in the study and a deeper explanation of the analytical tools used, particularly the TOPSIS method.

Additional justification for the choice of countries revised (page 6, first part of paragraph 2).

Additional explanation of TOPSIS revised (page 6, second part of paragraph 2).

  1. Results: The results are clearly presented, but further discussion on how these findings relate to broader trends in e-government development in Africa would be beneficial.

Findings related to broader trends in e-government development in Africa were revised (pages 11 &12).  

  1. Discussion: The discussion makes good connections with existing literature. It could be strengthened by more explicitly linking the results to the implications for future e-government strategies in Ghana and other African Union member states.

Implications for future e-government strategies in Ghana revised (page 15)

Round 2

Reviewer 1 Report

Comments and Suggestions for Authors

The authors have eliminated a certain number of shortcomings. Unfortunately, there are still some shortcomings that I think should be eliminated:

- After expanding the literature, the introduction became too long. I think that the introduction and literature analysis should be separated.

- A model flowchart is missing. Take a look at some other works to see what the flowchart model should look like. It is a figure with steps and phases of the model - Algorithm. For example see the paper: Tešić, D., & Marinković, D. Application of fermatean fuzzy weight operators and MCDM model DIBR-DIBR II-NWBM-BM for efficiency-based selection of a complex combat system. Journal of Decision Analytics and Intelligent Computing 2023, 3(1), 243–256.

- Check equations (1) and (4). Consult multiple sources of literature. For example see the paper: Liu, L.; Wan, X.; Li, J.; Wang, W.; Gao, Z. An Improved Entropy-Weighted Topsis Method for Decision-Level Fusion Evaluation System of Multi-Source Data. Sensors 2022, 22, 6391

- Describe the meaning of all the symbols used in the equations.

- The symbol Hj for max(wij) in equation (7) is not good because it is the same as one of the symbol in the initial decision matrix (line 319).

- The authors did not specify that all criteria are benefit type. If there is a cost criterion, then equation 6 is not good.

- In the last round I made this remark to the authors: „A sensitivity analysis is missing. A sensitivity analysis should be done using the change in weighting coefficients. Considering that the Entropy method was used to define the weighting coefficients, the sensitivity analysis is very important in the research.“ I still think that when using the entropy method it is very important to do a sensitivity analysis. Although this method belongs to the so-called group of objective methods, this does not mean that the results are objective. This is information that tells us that the attitude of man is excluded when defining weight coefficients. In practice, it happens that the method does not give good results. Therefore, the stability of the output results when changing the weight coefficients of the criteria is very important. Sensitivity example authors can see in the paper: Tešić, D., & Marinković, D. Application of fermatean fuzzy weight operators and MCDM model DIBR-DIBR II-NWBM-BM for efficiency-based selection of a complex combat system. Journal of Decision Analytics and Intelligent Computing 2023, 3(1), 243–256.

- I ask the authors to send me the calculations they did so that I could perform an additional control of the calculations.

Author Response

Authors responses to review comments

We appreciate your comments, below are our responses based on what we can do without it affecting other review comments.  

- After expanding the literature, the introduction became too long. I think that the introduction and literature analysis should be separated.

The introduction has been revised in reference to other Sustainability papers.

- A model flowchart is missing. Take a look at some other works to see what the flowchart model should look like. It is a figure with steps and phases of the model - Algorithm.

Based on your experts’ advice, we have modified this work by including the requested model algorithm. Kindly see Figure 1 on page 7.

- Check equations (1) and (4). Consult multiple sources of literature.

We are very grateful for your experts’ suggestion. Equations (1) and (4) have been checked.

- Describe the meaning of all the symbols used in the equations.

We thank you so much for your comment. Respectfully, all the symbols used in our equations, (1)-(8) are well explained as contained in the paper.

- The symbol Hj for max (wij) in equation (7) is not good because it is the same as one of the symbol in the initial decision matrix (line 319).

We appreciate your valuable suggestion. The concern raised here about the symbol in (7), has been worked on.

- The authors did not specify that all criteria are benefit type. If there is a cost criterion, then equation 6 is not good.

Thank you for your experts’ observation. We gave specification on page 8. Moreover, the condition mentioned was satisfied, with Max weight values are seen as (+) while the Min are (-)

- In the last round I made this remark to the authors: „A sensitivity analysis is missing. A sensitivity analysis should be done using the change in weighting coefficients. Considering that the Entropy method was used to define the weighting coefficients, the sensitivity analysis is very important in the research.“ I still think that when using the entropy method it is very important to do a sensitivity analysis. Although this method belongs to the so-called group of objective methods, this does not mean that the results are objective. This is information that tells us that the attitude of man is excluded when defining weight coefficients. In practice, it happens that the method does not give good results. Therefore, the stability of the output results when changing the weight coefficients of the criteria is very important.

We appreciate your valuable suggestion so much. Nonetheless, the robustness of our model is of no doubt since as you rightly pointed out, there was no human interference in determining the criteria weighting coefficients. Beside, considering papers that have similar features as ours from the literature, we observed that sensitivity analysis is not that needed anytime weighting coefficients are determined through scientific procedure such as the Entropy. Respectfully, permit as to mention a few of such papers found in the literature; Zhao, D.-Y., Ma, Y.-Y., & Lin, H.-L. (2022). Using the Entropy and TOPSIS Models to Evaluate Sustainable Development of Islands: A Case in China, Sustainability; Ardielli, E. (2019). Use of TOPSIS Method for Assessing of Good Governance in European Union Countries. Review of Economic Perspectives

 Moreover, we believe that in an attempt to obtain further results which may arguably not have any significant impact on the current findings may lead to a fresh review by all reviewers.

- I ask the authors to send me the calculations they did so that I could perform an additional control of the calculations.

Respectfully, we showed in Table 1 the Compiled Data for the 16 variants and 6 criteria used for this paper.

Round 3

Reviewer 1 Report

Comments and Suggestions for Authors

The authors removed some of the objections. However, not all objections have been removed:

- The authors did not specify that all criteria are benefit types. If there is a cost criterion, then equation 6 is not good. - The authors claim in their latest response that this data is on 8 pages. Unfortunately, I did not find them there. I ask the authors to clearly enter information about the type of criteria (cost or benefit) next to the description of the criteria. Without this data, readers who want to check the calculation cannot do so.

- In the last round I made this remark to the authors: A sensitivity analysis is missing. A sensitivity analysis should be done using the change in weighting coefficients. Considering that the Entropy method was used to define the weighting coefficients, the sensitivity analysis is very important in the research. I still think that when using the entropy method it is very important to do a sensitivity analysis. Although this method belongs to the so-called group of objective methods, this does not mean that the results are objective. This is information that tells us that the attitude of man is excluded when defining weight coefficients. In practice, it happens that the method does not give good results. Therefore, the stability of the output results when changing the weight coefficients of the criteria is very important. - For this remark, the authors stated that it is not necessary to address it and that a sensitivity analysis is not required. I do not want to open a discussion with the authors through the review process. By referring to papers, one can speak in favor of both opinions. I do not doubt that a sensitivity analysis is needed, and not only a sensitivity analysis through a change in weighting coefficients, but also an analysis of a change in the type of criteria, measurement unit, and the like. However, I know that a large part of reviewers do not insist on sensitivity analysis and that in a large number of papers (related to MCDM), there is no sensitivity analysis. That's why I give up this note.

Author Response

Comments and Suggestions for Authors: Reviewer 1 R3

- The authors did not specify that all criteria are benefit types. If there is a cost criterion, then equation 6 is not good. - The authors claim in their latest response that this data is on 8 pages. Unfortunately, I did not find them there. I ask the authors to clearly enter information about the type of criteria (cost or benefit) next to the description of the criteria. Without this data, readers who want to check the calculation cannot do so.

We thank you so much for your experts’ observations and advice. Respectfully, what we meant on “page 8” was about the indication that the model has both benefit and cost indexes. Nonetheless in this revised version, we have carefully studied the concern raised regarding equation (6) and have modified it to curtail both the benefit and cost indexes.

Thank you for your remark on this subject matter of our paper and for bringing a finality to it.  We appreciate your tremendous contribution in reshaping our paper for final publication.